# A New Grating Thermography for Nondestructive Detection of Cracks in Coatings: Fundamental Principle

**Zhi Qu [1], Weixu Zhang [1,2,*], Zhichao Lv [1] and Feng Wang [1]**

[1]  State Key Laboratory for Strength and Vibration of Mechanical Structures, Xi'an Jiaotong University, Xi'an 710049, China

[2]  John A. Paulson School of Engineering and Applied Sciences, Harvard University, Cambridge, MA 02138, USA

*  Correspondence: zhangwx@mail.xjtu.edu.cn; Tel.: +86-29-8266-8754

**Abstract:** It is important to detect the surface and/or subsurface cracks in coatings because the cracks usually indicate the failure of the system. Conventional detection techniques face two main challenges. One is the locating of the shallow cracks or defects in thin coatings. The other is the detection of the vertical cracks. Conventional infrared thermography can efficiently detect the horizontal cracks or defects. However, when locating the shallow cracks, it requires a high sampling frequency which is unrealistic for most of the infrared cameras. In terms of the vertical cracks, it is invalid since the propagation of its detecting signal is parallel to the cracks and does not interact with them. We introduce a new grating thermography method to overcome the two difficulties. In this paper we mainly illustrate its fundamental principle, which is validated by numerical simulations and a simple experiment. Overall, the principle analysis shows that grating thermography is highly effective in detecting cracks in coatings.

**Keywords:** infrared thermal wave; cracks; nondestructive detection; grating; coatings

## 1. Introduction

The detection of surface cracks has always been a key issue in structure safety and reliability in thermal barrier coatings [1–4], thin films, and other coating-like structures [5–8]. Surface cracks, including horizontal cracks and vertical cracks, significantly threaten the safety and reliability of coatings. They are usually generated by stress, thermal mismatch, or chemical circulation [5]. The extension of horizontal cracks causes the delamination of coatings [1,2]. The vertical cracks in coatings can extend to the interior of a structure and lead to its failure [2–4]. To avoid further disasters caused by surface cracks and to prevent their potential risk, the detection and location of surface cracks are key preconditions.

The detection of surface cracks in coatings faces many difficulties. First, the vertical cracks in coatings are difficult to detect because in most detection methods the propagation of the detecting signal is parallel to the vertical cracks and does not interact with them. The traditional methods, such as lock-in infrared thermography [8], pulsed infrared thermography, penetrant testing, and so on [9], are thus often invalid or insensitive to detect the vertical cracks. The eddy current testing method does detect the vertical cracks in conduction [7], but it is invalid for insulators [6]. Second, although it is relatively easy to find the horizontal cracks in coatings, the accurate locating of the cracks, however, remains a huge challenge. For example, in multi-layer coatings, the cracks may exist in different layers or interfaces. Due to the small thickness of each layer, to locate the position of cracks requires high resolution of signals, especially for coatings less than a few hundred micrometers. Third, due to the

large number of the cracks in coatings, the efficiency of detecting these cracks is another focus of research. Most existing Non-Destructive Testing (NDT) methods are designed to detect cracks one by one, such as eddy current detection [7] and ultrasonic testing [8]. Therefore, these methods require a relatively long time to detect all the cracks in coatings. Currently, the detection of surface cracks in coatings is still a research hot spot in the nondestructive testing field [10–16].

Using the detector array of an infrared camera, infrared thermal wave detection is an effective method to detect massive cracks which are parallel to the surface. Thermal waves are formed on the surface by absorption of modulated radiation and propagate towards the inside of the sample [13]. When they meet cracks or defects, the interaction between the thermal wave and the crack generates a new thermal wave signal which will propagate to the surface. Then, the response signal can be detected by an infrared camera [12]. In this method the detector array can simultaneously detect multiple horizontal cracks. The recently developed lock-in thermography combines infrared thermal wave imaging technology with the digital phase-locked loop signal processing technology [13–18]. It determines the defect characteristics by calculating the amplitude and phase diagram of the sample temperature at each point on the surface. It has high signal-to-noise ratio and has been widely used in the fields of nondestructive testing and evaluation. The detection depth of thermal waves is inversely proportional to the temporal frequency. To locate the depth of horizontal cracks, a more efficient method, called frequency modulated thermal wave imaging (FMTWI), was introduced [14].

Although the previous infrared thermal wave methods are efficient for horizontal cracks, most of them are invalid or insensitive to the vertical cracks because the propagation of thermal waves is parallel to the vertical cracks and thus is not affected by the vertical cracks. Recently, some other emerging infrared thermal wave methods, such as vibrothermography [19] and laser-line (or laser-spot) thermography [20–23], have been developed and used to detect vertical cracks. The vibrothermography method uses a specific acoustic excitation source to produce mechanical vibration inside the structure, and the cracks produce energy depletion and release heat, which eventually causes the rise of local temperature. Vertical cracks can be detected by this method, but its acoustic excitation source can only detect specific cracks. Besides, it requires coupling materials and complex equipment, so its versatility is poor. The laser-line (or laser-spot) thermography also has some shortages. It needs a very long time to detect a large-area surface because they need to carefully scan the surface with a small single detection area [22]. Furthermore, to accurately locate the cracks or defects in very thin coatings, a high temporal frequency of thermal waves is required, as well as a high sampling frequency of the infrared camera, which is unrealistic for most of the infrared cameras [7–9,23–32].

To overcome the disadvantages of all the above methods and to detect horizontal and vertical cracks simultaneously and efficiently, we introduce a new infrared grating thermal wave imaging method. This method can detect both the vertical cracks and the horizontal cracks simultaneously. Moreover, when detecting the shallow cracks or defects in coatings, it does not need a high sampling frequency of the infrared camera. In addition, it is highly efficient and easy to use. Same as the traditional thermal wave methods, the thermal waves cover the whole surface of coatings and all the cracks can be detected simultaneously in the new method. Furthermore, the lock-in technique can be used directly in our method. Our detecting method is of high academic importance and can greatly benefit the detection of surface cracks in practice.

In this paper, we mainly focus on the theoretical analysis of the propagation of the grating thermal wave to provide a clear physical explanation of the detecting principle, which gives a theoretical foundation of thermal wave detection. Then, numerical simulations and a simple experimental test were carried out to validate this method.

## 2. Theoretical Analysis

In the grating thermography, in order to detect the vertical cracks and the horizontal cracks, we project a series of striped moving light formed by a projector on the surface of the sample, i.e., a light grating with sinusoidal spatial intensity along the $x$ direction, as shown in Figure 1. The grating

wavelength l is the distance between two adjacent stripes of light. The incident light moves with a constant velocity along the $x$ direction. After some time, the temperature distribution, due to the injected grating light, reaches a steady state and it has the following form:

$$T(x, y, t) = T_0 \sin\left(\frac{2\pi}{l}x - 2\pi f t + \varphi_0\right) \tag{1}$$

where $f$ is the temporal frequency of the input signal; $l$ is the wavelength of the grating along the $x$ direction at the surface; and $\varphi_0$ is the initial phase angle and is a constant. From Equation (1) we can see that this is a moving thermal wave along $x$ direction with a velocity $v = f \times l$. We can control the grating wavelength by adjusting the distance of the light gratings through the projector. We can also control the temporal frequency $f$ by changing the moving speed $v$ of the light grating under a fixed grating wavelength $l$. The thermal wave propagates not only along the $y$ direction, but also along the $x$ direction. When it meets a crack, a reflecting thermal wave will propagate to the surface of the sample. The output signal of thermal waves can be detected by an infrared camera. By adjusting the temporal frequency and moving velocity of the illuminating light in $x$ direction, both the vertical cracks and horizontal cracks can be detected and located.

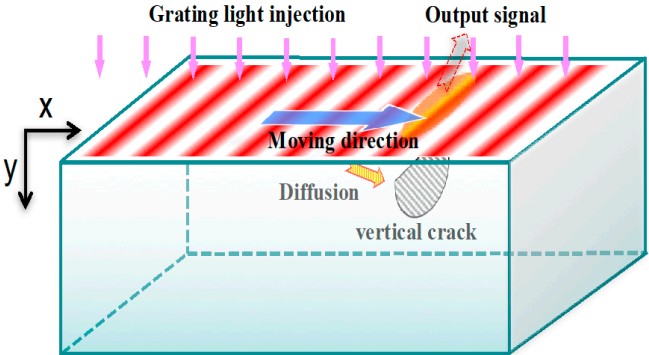

**Figure 1.** Illustration of infrared grating thermal wave imaging method. The red stripes represent the light gratings, they move from left to right along $x$ direction. The light gratings generate heat flux at the surface and thus thermal waves form at the surface correspondingly. When the thermal waves meet cracks or defects, they will be reflected and will further propagate to the surface. By monitoring the temperature signals of the surface, the cracks can be detected.

From the heat conduction equation, we derive a general solution of the grating thermal wave. The Cartesian coordinate system is shown in Figure 1. The coordinate origin is on the surface of the sample. The $x$ coordinate is in the horizontal direction, while the $y$ coordinate is in the vertical direction and points to the inside of the sample. According to the first law of thermodynamics and the Fourier law of thermal conduction, which assume that the object is homogeneous, isotropic, and does not have an inner heat source, the two-dimensional transient heat conduction partial differential equation is as follows:

$$\frac{\partial^2 T(x, y, t)}{\partial x^2} + \frac{\partial^2 T(x, y, t)}{\partial y^2} = \frac{1}{\alpha}\frac{\partial T(x, y, t)}{\partial t} \tag{2}$$

where $T(x, y, t)$ is the temperature; $\alpha = \sigma/(\rho c)$ is the thermal diffusivity; $\rho$ is the density; $c$ is the specific heat; and $\sigma$ is the thermal conductivity.

To obtain the theoretical solution of the grating thermal wave, we assume that the thickness of the sample is infinite. As shown in Figure 1, the moving grating light generates a thermal wave on the surface. Then the thermal wave, like a damping wave, propagates with energy absorption in a solid. For an infinite large half space without cracks, due to the geometric property, the decay of thermal waves only depends on the depth. When $y\rightarrow\infty$, the amplitude of the thermal waves will be zero.

Note that we only consider the steady-state thermal wave and neglect the initial transient response. Accordingly, we obtain the steady-state solution of thermal waves from Equation (1) as follows:

$$T(x,y,t) = T_0 e^{-y/\eta} [\cos 2\pi(\frac{y}{\kappa} + \frac{x}{l} - ft) + i \sin 2\pi(\frac{y}{\kappa} + \frac{x}{l} - ft)] \tag{3}$$

where $T_0$ is the amplitude of thermal wave at the surface; $\eta$ is the length scale of thermal wave diffusing into the solid, i.e., skin depth; $\kappa$ is the wavelength of thermal wave in the y direction; $f$ is the temporal frequency of the input signal; and $l$ is the grating wavelength in $x$ direction as those in Equation (1). $f$ and $l$ are the input parameters. $\eta$ and $\kappa$ are the response parameters of the thermal wave. After a tedious derivation, we find that these parameters satisfy the following relation:

$$\frac{1}{\eta} = \sqrt{2}\pi \sqrt{\frac{1}{l^2} + \sqrt{\frac{1}{l^4} + \frac{f^2}{(2\pi\alpha)^2}}} \tag{4}$$

$$\frac{1}{\kappa} = \frac{1}{\sqrt{2}} \sqrt{-\frac{1}{l^2} + \sqrt{\frac{1}{l^4} + \frac{f^2}{(2\pi\alpha)^2}}} \tag{5}$$

The above steady-state solution is also a theoretical foundation to analyze other thermal wave methods. For example, the thermal wave of the scanning laser-line thermography can be expressed into Fourier series, and the response of the single frequency is as Equation (3). The overall response of the scanning laser-line thermography is the linear superposition of a series of thermal waves.

When $l \to \infty$, our theoretical solutions, i.e., Equations (3)–(5), degenerate into the classical ones, which are corresponding to the conventional frequency modulated method (lock-in infrared thermography) [13], as follows:

$$T(y,t) = T_0 \mathrm{e}^{-y/\eta_0} \cos 2\pi\left(\frac{y}{\kappa} - ft\right) \tag{6}$$

$$\frac{1}{\eta_0} = \sqrt{\frac{\pi f}{\alpha}} \tag{7}$$

As shown in Equation (6), the heat flows only in the y direction and $\partial T(x,y,t)/\partial x = 0$. Since no heat flows in the $x$ direction, the thermal wave will not interact with the vertical cracks and the conventional methods thus cannot detect the vertical cracks. While in our method, shown from Equation (3), we can see that the heat also flows in the x direction, and the thermal wave can thus detect the vertical cracks. In addition, by comparing Equations (4) and (7), we can see that $\eta < \eta_0$, which means the thermal waves of our method have a smaller diffusion depth.

From Equation (4), it can be seen that the skin depth $\eta$ is a function of the temporal frequency $f$ and the grating wavelength $l$. When $l \to \infty$, our solution degenerates into that of the conventional thermal wave method and $\eta$ degenerates into the skin depth of conventional thermal wave method $\eta_0 = \sqrt{\alpha/\pi f}$. In the conventional method, the skin depth $\eta_0$ is only inversely proportional to $f$. Figure 2 shows the dependence of the skin depth $\eta$ on the grating wavelength $l$. With the decrease of $l$, especially when $l < 10\eta_0$, the skin depth $\eta$ decreases significantly.

In our method, the skin depth $\eta$ is an important parameter that determines the depth of detection. In order to detect the cracks deeply below the surface, we should let the skin depth be large enough. When detecting the shallow surface cracks, we should let the skin depth to be small enough to avoid the noise from deep inside the sample. For a very thin film, it is valuable to locate the shallow surface cracks and then to identify the origin of the crack. For a multi-layer thin film, identifying the location of a crack and thus distinguishing which layer is delaminated are also very important. In both cases, a small skin depth is needed. In conventional frequency modulated thermal wave methods, the only way to produce a small skin depth is to use a high temporal frequency $f$. However, it is very difficult

to collect the signal of high frequency thermal wave due to the low sampling frequency of infrared cameras. By contrast, in our new method, the high temporal frequency *f* is not necessary. We can use a short grating wavelength *l* and a low temporal frequency *f* to generate a small skin depth, which significantly decreases the requirement for the infrared camera. We believe that this is one of the main advantages of our new detection method.

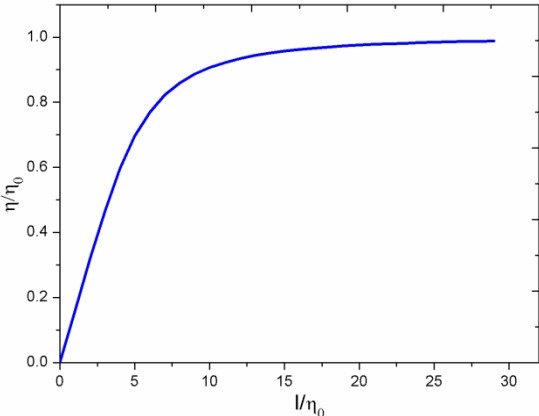

**Figure 2.** The dependence of skin depth ($\eta/\eta_0$) on the grating wavelength ($l/\eta_0$).

The wavelength $\kappa$ in y direction is another parameter of the thermal wave. When $l \to \infty$, it decreases to the conventional result $\kappa = \kappa_0 = 2\sqrt{\pi\alpha/f}$, and $\kappa$ will increase with the decrease of *l*, as shown in Figure 3. When the grating wavelength *l* is small, due to the fast attenuation of the thermal wave and the large wavelength in the y direction, the temperature distribution in the y direction is highly similar to an exponential function than a wave function. In the limiting case, when time frequency *f* is 0, the $\kappa$ is infinity, and the skin depth is completely determined by *l*, i.e., $\eta = l/2\pi$. In such a case, by adjusting the grating wavelength *l*, we can locate the position of the cracks. Still, the temperature distribution is sensitive to the vertical cracks. By comparing the visible light grating with the infrared thermography, the cracks can be detected.

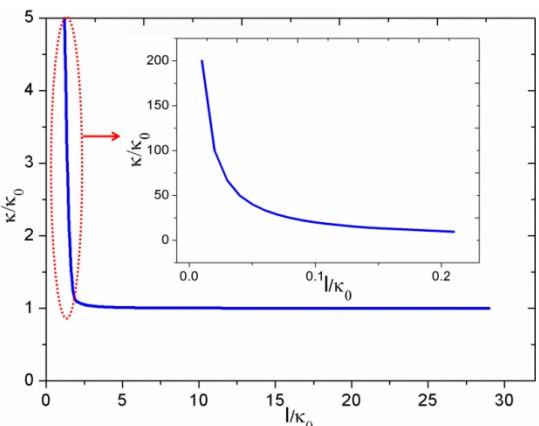

**Figure 3.** Effect of grating wavelength ($l/\kappa_0$) on the wavelength of thermal wave in *y* direction ($\kappa/\kappa_0$). The inserted image is the enlargement of the selected portion.

The analytical solution Equation (3) helps us understand the propagation rule of the thermal waves. When the thermal waves are generated at the surface of a sample through a moving grating light, their propagation in the sample can be fully described by the skin depth $\eta$ and the wavelength $\kappa$. When they meet cracks or defects, they will be reflected, scattered and/or refracted. The response signals of cracks also depend on $\eta$ and $\kappa$. When the response signal reaches the surface, it will influence

the temperature of the surface, thus the cracks can be detected. If we want to detect a deep crack, the skin depth η should be large enough, which means we should use a larger grating wavelength *l* and/or a lower temporal frequency *f*. If we want to detect a shallow crack, the skin depth η should be small enough to avoid the interference of the signal from deep cracks, so we should use a smaller grating wavelength *l* and/or a higher temporal frequency *f*. Then, different η and κ can be used to locate the cracks at different depths. In what follows we use numerical simulation to validate this principle.

## 3. Numerical Simulation

Applying our method to detect cracks or defects, a grating light is firstly projected to the surface and induces heat flux, which is a transient process. The surface temperature includes an initial transient response and a steady-state thermal wave response. In this paper, we only consider the steady-state thermal wave response and neglect the initial transient response. The surface cracks influence both the amplitude and the phase angle of thermal waves, but cannot influence the temporal frequency. The solution we gave, Equation (3), is a complex general solution. The real thermal wave can be expressed by a combination of a series of complex solutions.

When the thermal response reaches the steady state, the temperature signal of each point on the surface will be $A\sin(-2\pi ft + \varphi) + B$, where A is the amplitude, $\varphi$ is the phase angle, and B is the average temperature. At each point on the surface, $A$, $\varphi$, and $B$ can be fitted by the change of temperature signal with time. When there is no crack or defect, the phase angle will be a linear function of $x$, i.e., $\varphi = 2\pi x/l + \varphi_0$, where $\varphi_0$ is a constant.

In order to analyze the temperature distribution caused by the heat flow on the surface of the specimen, we consider a 2D case and use the numerical method to simulate the process. It is meaningful because it can not only exclude the influence of complex conditions such as noise or uneven heating in the real infrared nondestructive testing process, but also show clearly the contribution of parameters to the detection sensitivity.

Here, the difference method is adopted to calculate the 2D thermal wave problem. When establishing the difference scheme, we make Equation (2) discrete, making the discretization of area $0 \le x \le L$ and $0 \le y \le L$ with the node ($I$ =1, 2, … $p$+1) and ($J$ =1, 2, … $q$+1), as shown in Figure 4. The distance between the two nodes is called "distance step". Meanwhile, we disperse the time area $t \ge 0$ with the note $n$ ($n$ = 1, 2, … ). The interval between the two time points is called "time step". In the calculation of each time point, we presuppose a temperature field, of which the temperature field of the former time point is used as the initial field of the present. We then use the iterative calculation until the results converge.

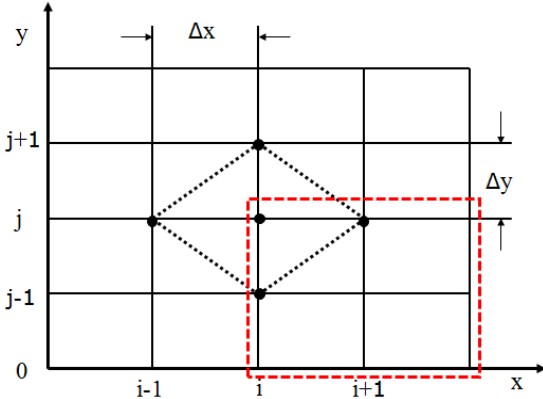

**Figure 4.** Discrete mesh partition of the heat conduction equation.

Making the finite difference of the heat conduction Equation (2), the discrete form is written as the following:

$$\left(\frac{\partial^2 T}{\partial x^2}\right)^{n+1}_{(i,j)} + \left(\frac{\partial^2 T}{\partial y^2}\right)^{n+1}_{(i,j)} = \frac{1}{\alpha}\left(\frac{\partial T}{\partial t}\right)^{n}_{(i,j)} \tag{8}$$

Expressing the left side in the central difference quotient and the right side in the forward difference quotient, we obtain:

$$\frac{T^{n+1}_{i-1,j} - 2T^{n+1}_{i,j} + T^{n+1}_{i+1,j}}{\Delta x^2} + \frac{T^{n+1}_{i,j-1} - 2T^{n+1}_{i,j} + T^{n+1}_{i,j+1}}{\Delta y^2} = \frac{1}{\alpha}\frac{T^{n+1}_{i,j} - T^{n}_{i,j}}{\Delta t} \tag{9}$$

The truncation error is: $o\left(\Delta t^2 + \Delta x^2 + \Delta y^2\right)$, which can be ignored. Let $C = \alpha\Delta t^2/\left(\Delta x^2 + \Delta y^2\right)$ and the convergence condition is $C \le 1/4$.

For each point inside the grid area, let $\Delta x = \Delta y$, and substitute $C$ into Equation (9), then temperature can be calculated based on the following equation:

$$T^{n+1}_{i,j} = C(T^{n}_{i,j-1} + T^{n}_{i,j+1} + T^{n}_{i+1,j} + T^{n}_{i-1,j}) + (1 - 4C)T^{n}_{i,j} \tag{10}$$

In numerical validation, we chose a sample with a 10 cm × 10 cm size. In the sample, there is a horizontal crack with length 0.4 cm (from 4.8 to 5.2 cm in $x$ direction) parallel to the surface and it is 0.5 cm above the lower surface. When a beam of light is projected onto the lower surface, it is equivalent to a heat flux injected into the surface. As for the moving grating light, the expression of excitation source signal is a moving heat flux as indicated in Figure 1: $A\sin\left(\frac{2\pi}{l}\cdot x - 2\pi ft + \varphi_0\right)$, $A$ is 100 W/m². The boundary condition is that all the three sides except the heating surface are adiabatic. In the calculations, for the boundary condition applied please refer to Reference [33]. The sample is a kind of composite material. The physical parameters of the material are: The density is 1600 kg/m³, the specific heat capacity is 1200 J/(kg·K) at constant pressure, and the thermal conductivity is 5 W/(m·K). We assume the crack is adiabatic, so the heat cannot conduct through the cracks. The temporal frequency is 0.01 Hz, and grating wavelength $l$ is $4\pi$ cm. The number of grids $N$ is 100. We calculated the 2D heat transfer equation, and obtained the temperature distribution of the surface, which is the output signal. Note that after the response of the thermal wave reached the steady state, we then used the four-parameter fitting method to fit the thermal wave signal at each point as $A'\sin(2\pi f'\cdot t + \varphi') + B'$, where $A'$ is the amplitude of the output signal, $\varphi'$ is the phase of the output signal, and $B'$ is the average temperature of each point at the surface.

Firstly, we validate our new method by detecting the horizontal crack. Figure 5 shows the amplitude and phase signals of the output signal. It can be seen that the horizontal crack alters both the amplitude and phase of thermal waves. In terms of horizontal crack detection, the grating thermography method has the same ability as the traditional method (lock-in thermography). The comparison between the two methods is based on the same condition, using the same heating intensity and time frequency to detect the same object.

Now we discuss the interaction between the horizontal crack and the thermal wave through the temperature distribution. At a certain time when the thermal fluctuation of the surface reaches the steady state, the temperature distributions of both the new method and the traditional method are shown in Figure 6. Due to the horizontal crack impeding the heat flow, it leads to a sudden change of the temperature around the crack and it also influences the temperature distribution of the surface as shown in the enlargements in Figure 6. So, from the temperature distribution of the surface, the crack can be located.

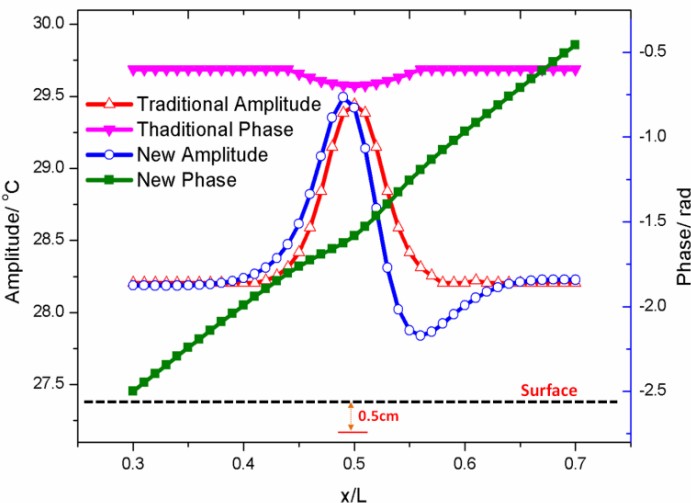

**Figure 5.** Comparison of amplitude and phase of detecting horizontal cracks. In the lower side of the diagram, the size and location of a horizontal crack is indicated as a red segment.

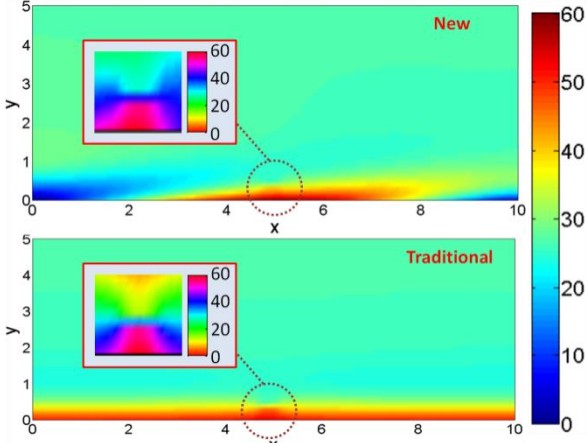

**Figure 6.** Comparison of temperature distribution (in °C) of detecting horizontal cracks. From the graph we can see that the crack impedes the heat flow, induces a sharp change of the temperature, and also leads to the variation of the surface temperature. Inside of each temperature nephogram, the enlargements of temperature distributions near the crack are given.

From the above simulation, it is shown that our method can detect the horizontal crack. Now we test its ability by detecting vertical cracks. The sample is also 10 cm × 10 cm. The vertical crack is at the center sample 0.2 cm below the surface, and its length is 0.6 cm. The other conditions are the same as that in the model with a horizontal crack. The temporal frequency is 0.01 Hz, and the grating wavelength is $4\pi$ cm. $A_0$ is the amplitude of the defect-free detection using the traditional method. As shown in Figure 7, the amplitude in our method shows a sudden change at the vertical crack position, which indicates that our method can effectively detect vertical cracks. However, there is no response in the amplitude of the traditional method. Above simulations validate that our method can detect both the horizontal cracks and the vertical cracks. The amplitude difference is the difference of amplitude between two adjacent points in $x$ direction and it can be calculated from the amplitude curve, as shown in Figure 7, which is helpful for quantitative studies in future.

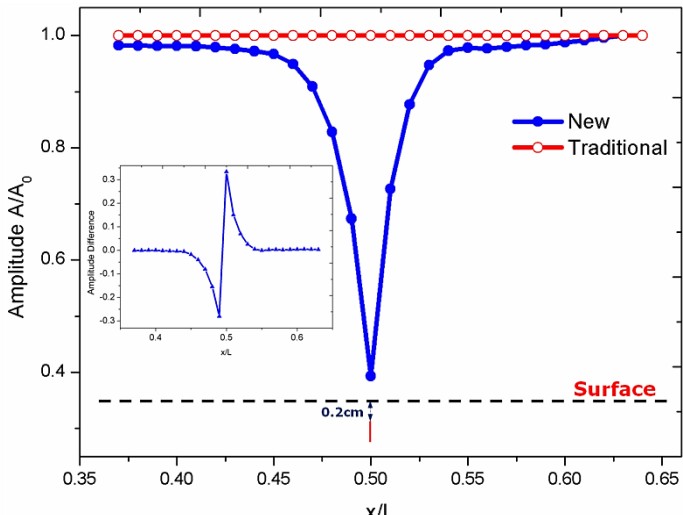

**Figure 7.** Comparison of test results of a vertical crack. In the lower side of the diagram, the size and location of a vertical crack is indicated as a red segment. In the left side of the graph, the amplitude difference of the new grating method is shown.

Then we checked the effect of heating parameters on detecting the horizontal cracks. The sample was the same as mentioned above. First, we considered the effect of temporal frequency at a fixed grating wavelength of $4\pi$ cm. As shown in Figure 8, the amplitude difference decreases with the increase of frequency and reaches 0 when the frequency is 0.1 Hz. This indicates that the frequency exerts a strong influence on the depth of heat conduction and the detection depth is very sensitive of temporal frequencies. Thus, the new method is especially suitable for detecting surface cracks.

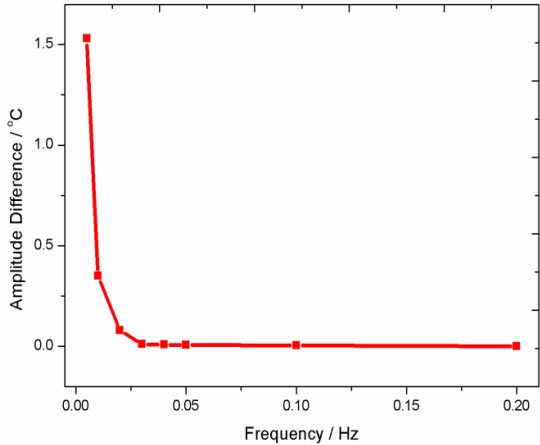

**Figure 8.** Variation of amplitude difference with temporal frequency.

Then we considered the effect of heating intensity. As shown in Figure 9, the amplitude difference, which is directly proportional to the heating intensity, increases only with the heating intensity. The relationship shows that, in order to improve the detection effectiveness, the signal of the crack can be enlarged by increasing the intensity of the excitation signal. The results obtained by numerical simulation can help us to improve the detection effectiveness in experiments. These numerical results are also consistent with our theoretical predictions.

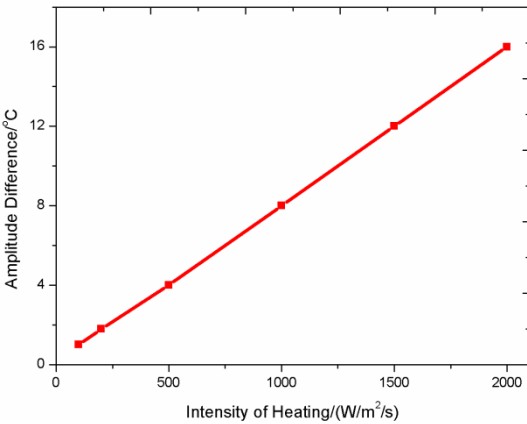

**Figure 9.** Variation of amplitude difference with heating intensity.

## 4. Experimental Validation

We built an experimental device to give a simple validation of our method. As shown in Figure 10, a typical infrared grating thermal wave testing system consists of a heating source, a data acquisition and processing system, and an infrared thermal camera. In our experiment, the infrared camera is a Nec R300 (NEC Avio Infrared Technologies Co., Ltd., Tokyo, Japan). The specimen was spliced by two 405 stainless steel blocks with the same shape. The two blocks had contact with each other. We wrapped a layer of paper whose thickness is 0.06 cm with weak reflections on the surface of the specimen to simulate the internal vertical crack at the spliced position. The stainless steel blocks were 10 cm × 4 cm ×1 cm. The temporal frequency was 0.02 Hz, and the grating wavelength was 40 cm. The heating source was a 750 W lamp projecting a grating thermal wave at equal intervals, and it was driven by the scanning drive system automatically to scan and heat the specimen.

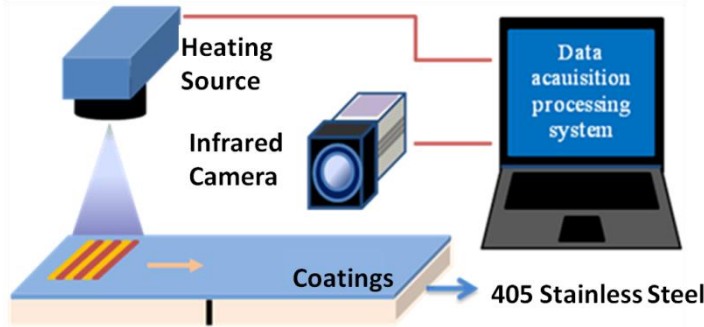

**Figure 10.** Infrared thermal wave testing system.

Figure 11 shows the amplitude distribution of the numerical simulation and the temperature distribution of the experiment (the inside color picture). From Figure 11, we can see the vertical crack has an obvious influence on the amplitude of the thermal wave. Therefore, the amplitude, arguably, can be used to detect the vertical cracks. Moreover, the experimental result proved that the temperature can also indicate the vertical crack. Therefore, the infrared grating thermal wave method is effective to detect vertical cracks. We calculated the corresponding amplitude difference as shown in Figure 11.

Note that the inside color picture is the measured temperature distribution of a sample in the experiment. The vertical crack can be easily figured out from the abrupt change of amplitude of the numerical results.

Overall, we proved that infrared grating thermography is an effective method to detect surface cracks in coatings. It not only contains all the advantages of the conventional thermal wave method, but also is effective to detect vertical cracks or defects. In principle, it has the advantage to locate the

cracks accurately. It is noted that in this paper we mainly illustrate the fundamental principle. In the future we will discuss the experimental technique further.

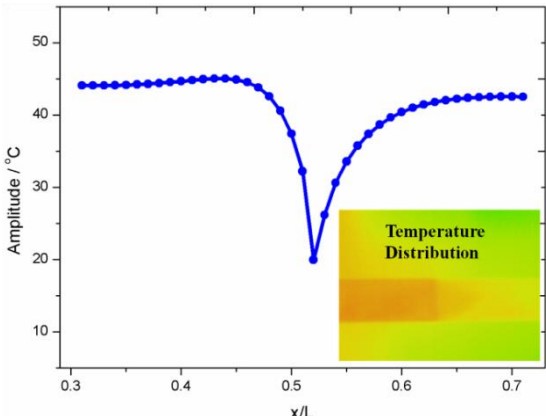

**Figure 11.** Numerical simulation and experimental results of infrared grating thermal wave imaging.

## 5. Conclusions

In summary, we clarified the fundamental principle of a new infrared grating thermography method which we introduced recently to detect the cracks in coatings. We presented the theoretical solutions to help understand the characteristic of thermal waves. The detection depth of grating thermography is determined by the grating wavelength and the temporal frequency of thermal waves simultaneously. The new method is effective to detect both the vertical cracks and the horizontal cracks. In principle, by varying the grating wavelength, the position of the cracks or defects can be located accurately. When locating the shallow cracks, it requires a low sampling frequency of infrared cameras. The principle was verified by the numerical simulations and the experimental results.

**Author Contributions:** Conceptualization, W.Z. and Z.Q.; Software, Z.Q.; Validation, W.Z., Z.Q. and F.W.; Formal Analysis, Z.Q. and Z.L.; Investigation, W.Z. and Z.Q.; Resources, Z.Q., Z.L. and F.W.; Data Curation, W.Z. and Z.Q.; Writing—Original Draft Preparation, Z.Q.; Writing—Review and Editing, W.Z.; Supervision, W.Z.; Project Administration, Z.Q.; Funding Acquisition, W.Z.

**Funding:** This work was supported by NSFC (Nos. 11772246, 11472203, 11172227), 973 Project (No. 2013CB035700) and was partially supported by New Century Excellent Talents in University (No. NCET-13-0466), and Natural Science Basic Research Plan in Shaanxi Province of China (No. 2013GY2-14).

**Acknowledgments:** Weixu Zhang acknowledges the support from China Scholarship Council. We very much appreciate Tony J. Zhang for his valuable discussion and suggestion in writing this letter.

**Conflicts of Interest:** The authors declare no conflict of interest.

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
