# Peer review of "A New Grating Thermography for Nondestructive Detection of Cracks in Coatings: Fundamental Principle"

_coatings, doi:10.3390/coatings9070411_

Round 1
Reviewer 1 Report
The paper describes an interesting method to detect cracks near the surface. It deserves to be published but a bigger effort to explain everything (equations, figures, procedures…) needs to be made. Following are the detailed comments:
1. English needs to be revised. Also, there are many typos throughout the text.
2. Page 5, line 106: Define temporal frequency. Include equations and figures to better describe the “moving light”. Again, in page 7, line 137: define with equations f and l.
3. Include references for equations 2.2, 2.3 and 2.4.
4. The first sentence of page 8 is difficult to understand. Please, rewrite.
5. Page 10, line 199: The heating source is described with an equation. This equation should appear at the beginning of section 2 defining and describing all the variables that appear in it.
6. Page 11, line 229; page 12, line231: when the authors say “layer”, do they mean “time step”? This explanation needs to be improved.
7. Does equation 3.3 come from equation 3.2? This should be explained.
8. Page 13, line 248: It seems that the authors are studying a vertical crack when they say that it is “perpendicular to the surface”. It was not until the next page that I realized it was a horizontal crack. The authors should clearly say at the beginning of page 13 that they are modelling a horizontal crack. On line 249 they say that more details are indicated in Figure 1 but in this figure there are no details.
9. Line 250: the expression in this line is the heat flux? In that case please explicitly say it. The units are in W/m^2/s. Is it a mistake or is this a flux rate? If that is the case I don’t understand why the excitation source is given in terms of flux rate.
10. Page 13, line 261: a crack does not have an amplitude and phase. This sentence has to be rewritten.
11. Figure 5: does the “phase” represent the delay between the excitation source and the temperature measured? Why is it not represented the “phase” in the traditional method?
12. Page 13, line 262: I don’t understand the conclusion at the end of the sentence “The area affected by the crack…”
13. Page 14, 270: what does “… the temperature distribution fluctuates apparently”?
14. Figure 6. A better explanation is needed for this figure. Is it representing the temperature distribution? There are actually two figures, I assume that the one on top for the new method and the other the traditional. But all these thing should be explained in the text and the figure caption.
15. Page 15, line 281: I think it is figure 7, not 6.
16. Page 15, line 285: the concepts “amplitude difference” and “amplitude curve” appear without an explanation of what they mean. “Amplitude difference” is defined in the following page but it is late and still not clear.
17. Page 17. Photographs of the experiments would be interesting and not just a scheme (figure 10).
18. Page 17, line 319: the authors say that they put a paper between two pieces of metal to simulate the crack but in figure 10 they show the paper over the surface of the two pieces but not between them. This needs to be clarified. A photograph would help.
19. Figure 11. Is it not possible to obtain the temperatures from the experimental measurement along a line so it can be drawn together with the one obtained from simulation?
Author Response
Dear professor:
First of all, let me express our gratitude for your serious, careful and helpful comments and suggestions on our work. These comments and suggestions have helped us to revise and improve our paper significantly. After carefully considering your comments, we change the title into “A New Grating Thermography for Nondestructive Detection of Cracks in Coatings: Fundamental Principle” to emphasize the key idea of our paper is the introduction of the basic principle of the new grating thermography. The principle and our analytical solutions have never been published elsewhere. We have made a major revision according to the comments, including, a more detailed introduction to the theory, the necessary supplements of the numerical simulation results, more careful discussions and more accurate conclusions. 3 Figures are redrawn. In addition, we also corrected some other inappropriate contents that we did not notice before. All the revised places are marked in red in the revision.
Thank you again.
Sincerely yours,
Weixu Zhang, Ph. D.
Professor,
State Key Laboratory for Strength and Vibration of Mechanical Structures,
Xi’an Jiaotong University,
No. 28, West Xianning Road, Xi’an, Shaanxi, China

Reviewer 2 Report
The research presented by Z. Qu et.al. presents an interesting and innovative approach for detection of both horizontal and vertical cracks in a variety of solid substrates. I think the manuscript's quality is good, but in my opinion it can be published only after serious revision (in order to end up with an article, which will attract as high as possible interest). I recognize a few major shortcomings, in particular:
1) The described method is indeed novel, in my opinion, however, after a brief literature research i see that the use of grating thermography has previously been suggested by other scientists - Z. Qu et.al., Research on finite element analysis of the infrared grating lock-in thermal imaging method, Int. J. Comp. Phys. Series (2018) 204-215. Furthermore, there are a few more relevant papers, which have been missed to be considered in the introduction. One of them is proposing vibrothermography, which might be used for vertical cracks detection - C. Xu, Experimental investigation on the detection of multiple surface cracks using vibrothermography with a low-power piezoceramic actuator, Sensors (2017) 2705.
2) More scientific input can be added in the description of the used numerical approach, as well as the materials.
3) The authors validate their method experimentally for the case of vertical cracks detection and location, but i don't see any verification (experimental) about the location of shallow horizontal defects.
4) Too many repetitions of particular phrases and words, also in some paragraphs the writing is not compelling enough, so the scientific content is hard to be assimilated completely.
More comments and suggestions can be find in the attached pdf file as "sticky notes".

Author Response

(The authors gave the same response as above.)

Round 2
Reviewer 1 Report
The paper has been improved and all the comments answered. I think it is ok now for publication.
Reviewer 2 Report
The authors have taken into account all my comments and indeed the manuscript's quality has improved significantly. I think it is ready to be published in Coatings. Just as a short note to the authors - the red strips in Figure 1 are missing again, probably still due to a pdf bug. What i suggest is to either fixing the bug (make sure it is fixed prior to publication) or adding the red strips using Paint. The latter might not be able to draw the strips as effectively as they appear in the revision cover letter, however, at least the Figure will be accurate.